# Effects of Different Types of Contraction Exercises on Shoulder Function and Muscle Strength in Patients with Adhesive Capsulitis

**DOI:** 10.3390/ijerph182413078

**Published:** 2021-12-11

**Authors:** Won-Moon Kim, Yong-Gon Seo, Yun-Jin Park, Han-Su Cho, Su-Ah Lee, Sang-Jun Jeon, Sang-Min Ji

**Affiliations:** 1Department of Sports Science, Dongguk University, Gyeongju-si 38066, Korea; kimwonmoon3426@hanmail.net; 2Samsung Medical Center, Department of Orthopedic Surgery, Division of Sports Medicine, Seoul 06351, Korea; 3Department of Health Rehabilitation, Osan University, Osan-si 18119, Korea; yjpark35@osan.ac.kr; 4Sports Medicine Center, Sunsoochon Hospital, Seoul 05556, Korea; suya16@hanmail.net; 5Department of Sports Science, Hanyang University, Ansan-si 15588, Korea; lko7774@naver.com; 6Parc Athletic Training Center, Gwangju-si 12767, Korea; sangjun4747@naver.com (S.-J.J.); sm96polo@hanmail.net (S.-M.J.)

**Keywords:** shoulder adhesive capsulitis, muscle contraction, exercise therapy, muscle strength, recovery of function

## Abstract

Although several studies have reported the effect of exercise therapy for adhesive capsulitis (AC), studies on the comparison of different exercise types on shoulder muscle strength and function in patients with AC are lacking. This study aimed to investigate the effect of different exercise types on shoulder muscle strength and function in patients with AC. Thirty female patients with AC were categorized into an eccentric contraction exercise group (ECG, *n* = 15; age, 51.53 ± 4.73 years) and a concentric contraction exercise group (CCG, *n* = 15; age, 52.40 ± 4.03 years). The participants in each group performed a different exercise program three times per week for 60 min per session for 12 weeks. The range of motion (ROM) of the shoulder joint, visual analog scale, shoulder muscle strength, and Constant–Murley score (CMS) were measured before the intervention and after 12 weeks of the exercise intervention. Shoulder ROM in flexion (increase of 31%) and external rotation (ER) (increase of 54%) showed a significant improvement in the ECG (*p* < 0.05). Muscle strength in ER was significantly different between the two groups (*p* < 0.05). Pain severity showed improvement in the ECG (decrease of 61%) after the intervention (*p* < 0.01). The CMS in the ECG (increase of 48%) showed a greater improvement than that in the CCG after the intervention (*p* < 0.01). This study showed that eccentric contraction exercise had a more beneficial effect than concentric contraction exercise for improving shoulder muscle strength and function in females with AC.

## 1. Introduction

Adhesive capsulitis (AC), also known as frozen shoulder, is characterized by shoulder pain, a decreased range of motion (ROM), and a decreased shoulder function [1]. The prevalence has been reported to range from 2% to 5% in the general population, and it is higher in women than in men [2,3]. The risk factors for AC have been reported to include chronic inflammation, endocrine and biochemical changes, diabetes mellitus, neurological factors, and long-term shoulder immobilization after surgery [4,5].

Several factors including a decreased ROM, muscle weakness, and the thickening of the joint capsule and synovial membrane are used for AC diagnosis [6]. In general, AC can be divided into three stages: freezing, frozen, and thawing [7]. The freezing stage is characterized by an increase in pain at night and usually lasts for 2 to 9 months. The next stage is the frozen stage and is referred to as a stiffness phase that is accompanied by a loss of shoulder motion and a decrease in pain. In general, this stage lasts between 4 and 12 months. The last stage has a gradual improvement of the shoulder motion and minimal pain. It is important to consider the AC stage and timing of the treatment for the early recovery of a frozen shoulder when applying an exercise intervention [8].

The first treatment option for AC is conservative management including medication, a local steroid injection, physical therapy, exercise therapy, hydrodistension, and manipulation under anesthesia [7,8,9]. Of the interventions, exercise therapy has been recommended as an intervention method for improving the shoulder ROM, muscle strength, and shoulder function [10,11,12,13,14]. The main types of exercise intervention are stretching, mobilization exercises, and strengthening exercises [8,10,12].

Several comparative studies on exercise interventions [1,15,16] have reported the differences on the shoulder ROM, pain, and function between two different interventions in patients with AC. One study [1] reported that a combined scapulothoracic exercise and glenohumeral exercise can be more effective in decreasing pain and increasing the ROM than only glenohumeral exercise. Guler-Uysal et al. [15] conducted a comparison on two different modes that included the Cyriax method and physical therapy, and reported that the Cyriax method provided faster and better effectiveness than the physical therapy in shoulder pain and the ROM. Another previous study [16] compared the intensity of mobilization exercises and reported that high-grade mobilization techniques were more effective than low-grade mobilization techniques in the shoulder ROM and function. Most comparative studies have focused on an analysis of the effect of combined exercises, the effect of different intensities, or the differences between exercise therapy and other interventions.

Muscle contraction including concentric contraction exercise (CCE) and eccentric contraction exercise (ECE) is a type of exercise mode and is characterized by a change in the muscle length [17,18]. Several comparative studies [19,20,21] have reported the results of a comparison between ECE and CCE in shoulder disease. One study [19] reported that ECE was better than CCE on pain (*p* = 0.006) and abductor strength (*p* < 0.001) in shoulder impingement syndrome. However, Dejaco et al. [20] reported that there was no significant difference between two different contraction exercises in rotator cuff tendinopathy. The study conducted by Blume et al. [21] reported that no difference was found on the shoulder function (*p* = 0.890), arm elevation active ROM (*p* = 0.373), and abductor strength (*p* = 0.421) for patients with shoulder impingement syndrome. The effect of ECE and CCE has been reported differently among shoulder diseases. 

AC, an inflammatory disease, is affected by shoulder pain and ROM more than other shoulder diseases. Several comparative studies [1,15,16] related to AC have been reported but these studies focused on the different effects of the two exercise types. Considering the pathophysiological differences between shoulder diseases, a comparative study on the two different contractile exercises is needed in patients with AC. Therefore, the purpose of this study was to compare the effectiveness of ECE and CCE on shoulder muscle strength and function in patients with AC.

## 2. Materials and Methods

### 2.1. Study Design

The sample size was calculated using an effect size of 0.25, α of 0.05, a power of 0.80%, and three measurements. A minimal sample size of 28 was calculated by the G-Power program, version 3.1.9.4 (Heinrich Heine University Düsseldorf, Düsseldorf, Germany) [22]. Thirty female patients diagnosed with AC participated in the study (Table 1). Thirty-four patients with AC participated; four patients withdrew for personal reasons. Thirty patients completed this study (Figure 1).

The diagnosis was conducted by an orthopedic surgeon with more than 15 years of experience. First, the patients were instructed to explain their symptoms and history; a physical examination was then conducted that included measurements of the shoulder ROM and a special test (Neer’s impingement test and Hawkins–Kennedy impingement test). The next step was to take an X-ray to distinguish from osteoarthritis. Finally, magnetic resonance imaging was performed for a final confirmation of AC. After the diagnosis, the patients who were diagnosed with primary AC (with no traumatic history and no osteoarthritis from the X-ray) were included in this study. The exclusion criteria were as follows: patients who had a history of surgery on the shoulder, other shoulder diseases including a rotator cuff tear, subacromial impingement syndrome, and a shoulder labral tear. All subjects who participated our study were in the freezing stage. The patients were randomly allocated according to the order of participation in the study into two groups: the ECE group (ECG; *n* = 15) and the CCE group (CCG; *n* = 15). 

All participants were informed of the study procedures and provided informed consent before participation. This study was approved by the Institutional Review Board (IRB) of DongGuk University (IRB No. DGU-20200028) and was conducted in accordance with the guidelines of the 1964 Declaration of Helsinki.

### 2.2. Exercise Program

In this study, the exercise program was set by modifying previous clinical practices [23,24]. Two groups performed the different types of muscle contraction; one group performed ECE and the other CCE. In the ECG, a concentric contraction was conducted for 2 s and then an eccentric contraction for 10–15 s. In the CCG, the time for the concentric contraction was the same but the eccentric contraction time was shortened to 3 s [18,19,20]. This exercise program was conducted three times per week for 60 min per session for 12 weeks. The main exercise time was 40 min; the other time was spent in warm-up and cool-down. The program was supervised by an exercise specialist with >10 years of clinical experience in orthopedic rehabilitation clinics. All subjects were not allowed to participate in any other exercise during the study period. The exercise intensity was monitored using a visual analog scale (VAS) with 3–4 points indicating mild pain. The explanation of the VAS was carried out using pictures and numbers. The participants had a resting time between each set of 30 and 50 s between exercises (Figure 2) (Table 2).

### 2.3. Outcome Assessments

#### 2.3.1. Constant–Murley Score (CMS) 

The CMS used in this study is one of the commonly used questionnaires to assess the shoulder function [25]. The CMS consists of four items: activities of daily living (20 points), pain (15 points), muscle strength (25 points), and ROM for joints (40 points) with a total score of 100 points; a lower score means a higher percentage of functional disability and a higher score means the opposite. The questionnaire has a relatively high accuracy and reliability (intra-class correlation coefficient = 0.86) [25]. In this study, the subject was instructed to fill out the questionnaire themselves.

#### 2.3.2. ROM Measurement

The active ROM in the shoulder joint was measured for forward flexion and abduction and the external rotation was measured using a plastic Baseline^®^ goniometer (model 12-100, New York, NY, USA) in a supine position with the knees flexed. The goniometer axis was placed over the center of the humeral head at the greater tuberosity. The fixed arm was placed along the side of the participant, which was aligned with the greater trochanter and was parallel to the floor. The moving arm was placed along the lateral aspect of the humeral shaft and aligned with the lateral epicondyle. For measuring the ROM of shoulder abduction, the goniometer axis was placed over the humeral head center at the greater tuberosity. The fixed arm was parallel to the sternum and the participant was instructed to move the arm in the direction of abduction. The ROM for the shoulder ER was measured at 90° elbow flexion and 90° shoulder abduction in the supine position. The goniometer axis was placed over the center of the olecranon process and styloid process of the ulna. The participants were instructed to move the arm in the direction of the ER. Each measurement was performed three times and the mean value was recorded [26] (Figure 3).

#### 2.3.3. Muscle Strength

A portable hand-held dynamometer (Power-Track II, J Tech Medical Industries, Midvale, UT, USA), which has a high reliability (ICC, r = 0.971–0.972) [27], was used to measure the muscle strength. For measuring the maximal strength of forward flexion, the forearm was pronated by 90° by extending the elbow joint in an anatomical position. To measure the external rotational strength of the shoulder joint, the measuring device was installed horizontally with the elbow joint flexed at 90° in a neutral sitting position and the back was fixed to the wall to prevent the movement of the trunk of the participant. The participant was instructed to pull toward the opposite direction for measuring the muscle strength during the contraction (Figure 4).

### 2.4. Statistical Methods

A descriptive analysis was performed to determine the mean and standard deviation. An independent sample *t*-test was performed to confirm the homogeneity of the baseline data between the groups and a normality test was performed using the Kolmogorov–Smirnov test. We confirmed that all variables in this study followed a normal distribution. An analysis of a two-way repeated measure ANOVA was performed to confirm the statistical differences between the groups and a paired *t*-test was conducted to confirm the differences between the time points in each group when showing significantly different interactions. The data analysis was performed using SPSS version 22.0 for Windows (IBM Corp., Armonk, NY, USA) and the significance of all data was set to *p* < 0.05.

## 3. Results

### 3.1. Shoulder Pain

Regarding the VAS for shoulder pain, a significant difference was found between the groups (*p* < 0.01, η^2^ = 0.301) and the interaction between the time points and the groups (*p* < 0.01, η^2^ = 0.274) as well as between the time points (*p* < 0.001, η^2^ = 0.964). A significant difference was observed after 12 weeks of intervention in both groups (*p* < 0.001, Cohen’s d = 10.999 and *p* < 0.001, Cohen’s d = 9.656, respectively) (Table 3).

### 3.2. Shoulder ROM

A significant difference was found in the ROM of flexion between the groups (*p* < 0.05, η^2^ = 0.177) and the interaction between the time points and groups (*p* < 0.05, η^2^ = 0.190) as well as between the time points (*p* < 0.001, η^2^ = 0.845). Both groups showed a significant difference in the ROM changes between the pre- and post-intervention (*p* < 0.001). In the ROM changes, a significant difference was noted only between the time points (*p* < 0.001, η^2^ = 0.962) in abduction. No significant difference was found between the groups (*p* = 0.200, η^2^ = 0.058) and the interaction between the time points and groups (*p* = 0.073, η^2^ = 0.110). Both groups showed a significant difference between the pre- and post-intervention (*p* < 0.001, Cohen’s d = −6.424 and *p* < 0.001, Cohen’s d = −3.332, respectively). In the ROM changes in the ER, a significant difference was noted between the time points (*p* < 0.001, η^2^ = 0.901) and between the groups (*p* < 0.05, η^2^ = 0.131). There was no significant difference in the interaction (*p* = 0.74, η^2^ = 0.109). A significant difference was observed after 12 weeks of intervention in both groups (*p* < 0.001, Cohen’s d = −5.975 and *p* < 0.001, Cohen’s d = −6.267, respectively) (Table 3).

### 3.3. Shoulder Muscle Strength

In the changes of muscle strength, there was a significant difference in the interaction between the time and the groups (*p* < 0.05, η^2^ = 0.177) and between the time (*p* < 0.001, η^2^ = 0.921) in shoulder flexion. However, no significant difference was noted between the groups (*p* = 0.230, η^2^ = 0.051). Both groups showed a significant difference between the pre- and post-intervention (*p* < 0.001, Cohen’s d = −5.160 and *p* < 0.001, Cohen’s d = −9.335, respectively). Regarding the changes in strength in the ER, a significant difference was noted between the two groups (*p* < 0.05, η^2^ = 0.144) and the interaction between the time and the groups (*p* < 0.001, η^2^ = 0.823) as well as between the time (*p* < 0.001, η^2^ = 0.967). A significant difference was seen after 12 weeks of intervention in both groups (*p* < 0.001, Cohen’s d = −12.114 and *p* < 0.001, Cohen’s d = −9.672, respectively) (Table 3).

### 3.4. Shoulder Function

For the changes in the CMS, a significant difference was found between the two groups (*p* < 0.01, η^2^ = 0.254) and the interaction between the time and the groups (*p* < 0.05, η^2^ = 0.169) as well as between the time (*p* < 0.001, η^2^ = 0.954). A significant difference was observed after 12 weeks of intervention in both groups (*p* < 0.001, Cohen’s d = −12.358 and *p* < 0.001, Cohen’s d = −7.046, respectively) (Table 3).

## 4. Discussion

This study aimed to compare two different exercises on the shoulder function and muscle strength in female patients with primary AC. The results of this study revealed that the ECG and CCG showed a significant improvement after 12 weeks of an exercise intervention on the shoulder ROM, pain, muscle strength, and function in patients with AC. Moreover, ECE appears to be more effective than CCE in improving the shoulder ROM, pain, muscle strength, and function.

According to the stages of AC, the disease has three phases of clinical presentation and the time applied for a proper intervention for reducing pain is very important in patients with AC [1,13]. In this study, the reduction of shoulder pain showed a significant difference after an exercise intervention of 12 weeks in both groups. However, the ECG showed a significantly greater improvement than the CCG. Several studies [19,21] on shoulder diseases such as subacromial impingement syndrome and rotator cuff tendonitis have reported that ECE is beneficial to improve pain, muscle strength, and function and shows a greater improvement in pain than CCE after an intervention [19]. Our results may explain that the early recovery of the shoulder ROM by ECE could reduce mechanical stress on the shoulder joint, resulting in an amelioration of shoulder pain. However, further studies with a larger sample size are needed to confirm this finding in patients with AC.

The therapeutic goals in AC are an increase of the passive and active ROM, pain reduction, and the improvement of the shoulder function [14]. Early recovery of the shoulder ROM is an important factor to consider for evaluating and predicting clinical outcomes. In the present study, the ROM of the shoulder joint in flexion and abduction as well as the ER significantly increased after the exercise intervention in both groups. However, in this study, there was a different finding that the ROM in flexion and the ER showed a greater increase in the ECG than in the CCG. This result was similar to the findings of a previous study by Chaconas et al. [19], which reported that TheraBand eccentric exercise was a beneficial intervention to improve shoulder muscle strength. Our findings contradicted the results of another study, which found no significant difference between ECE and CCE in patients with subacromial impingement syndrome [21]. The study reported no significant difference in the muscle function, active ROM, and strength. Previous studies [19,21] used a different method compared with our study, which was conducted over a longer eccentric period of 10–15 s. The eccentric contraction time performed in our study may have positively affected the vitrification of the adherent joint, resulting in an improved ROM in the shoulder joint [28]. Another possible mechanism may have been that the increased muscle length by ECE contributed to an increase in muscle tension, resulting in a greater increase in the ROM [29].

According to a systematic review, eccentric training appeared to be more effective at increasing muscle mass than concentric training in a healthy population [20]. ECE increases muscle strength and mass [18] and patients with subacromial pain syndrome showed a significant improvement after ECE [19]. Another study [21] reported that both eccentric and concentric programs improved the shoulder muscle strength in patients with subacromial impingement syndrome but no difference was found between the two exercise modes. In this study, both contraction exercise groups showed an improved shoulder muscle strength after the intervention but the ECG showed a greater increase than the CCG. ECE has an effective mechanical effort compared with CCE, resulting in increased muscle strength [30,31]. A systematic review also reported that eccentric training increases strength compared with concentric training [20]. Considering these aspects, we assumed that ECE increased the shoulder ROM and muscle activation through mechanical changes including the rearrangement of joint capsule collagen fibers and the vitrification of the adhered joint tissue [23].

The shoulder function in patients with AC is lower than that in the general population [15,23,24]. The CMS was previously used to evaluate the shoulder function of patients with a shoulder pathology [19,32]. The study showed that ECE was superior to CCE in patients with AC. A study comparing CCE and ECE in patients with subacromial pain syndrome demonstrated that the CMS changes were significantly greater in the ECG [19]. ECE increased the synovial fluid flow in the shoulder joint areas, softening the joint capsule tissue [28,29,33]. This indicated that that shoulder ROM and shoulder pain in the adhered joint tissue may have reduced the mechanical stress on the shoulder joint, resulting in an improvement of the shoulder function. However, further study is needed to compare the different questionnaires for evaluating the shoulder function of patients with AC.

This study has a few limitations. First, all study participants were females diagnosed with AC. Therefore, it is difficult to generalize the results for males with AC. Additional studies are needed to compare the differences between the sexes. Second, AC is divided into the following three stages: freezing, frozen, and thawing. The study participants were in the freezing stage and they had severe pain and a limited ROM in the shoulder joint. Therefore, the application of this result for patients in other stages is not suitable. Subsequent studies comparing patients in the frozen stage and those in the thawing stage are warranted.

## 5. Conclusions

ECE showed a more positive effect than CCE for the shoulder ROM, pain, muscle strength, and function in female patients with AC. Therefore, ECE should be considered as an essential component of exercise therapy programs for female AC patients. However, further investigations are necessary to identify the obvious mechanisms of the effects of eccentric training programs on the improvement of the shoulder function and muscle strength for patients with AC.

## Figures and Tables

**Figure 1 ijerph-18-13078-f001:**
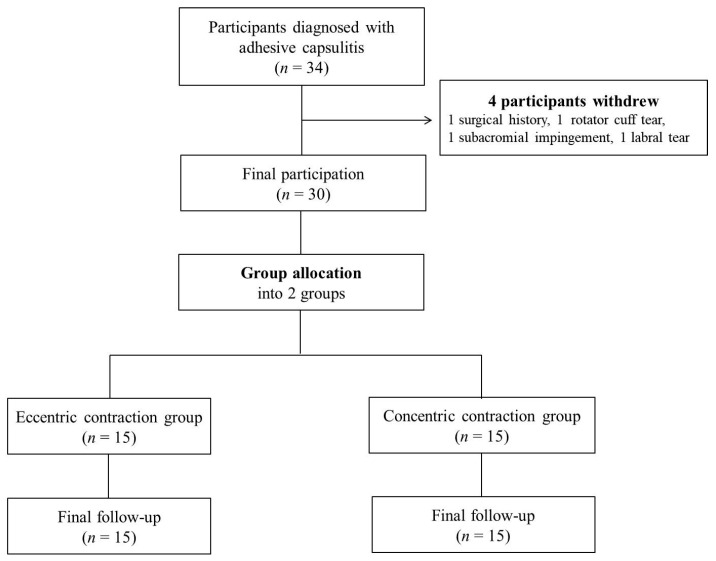
Flow chart of this study.

**Figure 2 ijerph-18-13078-f002:**
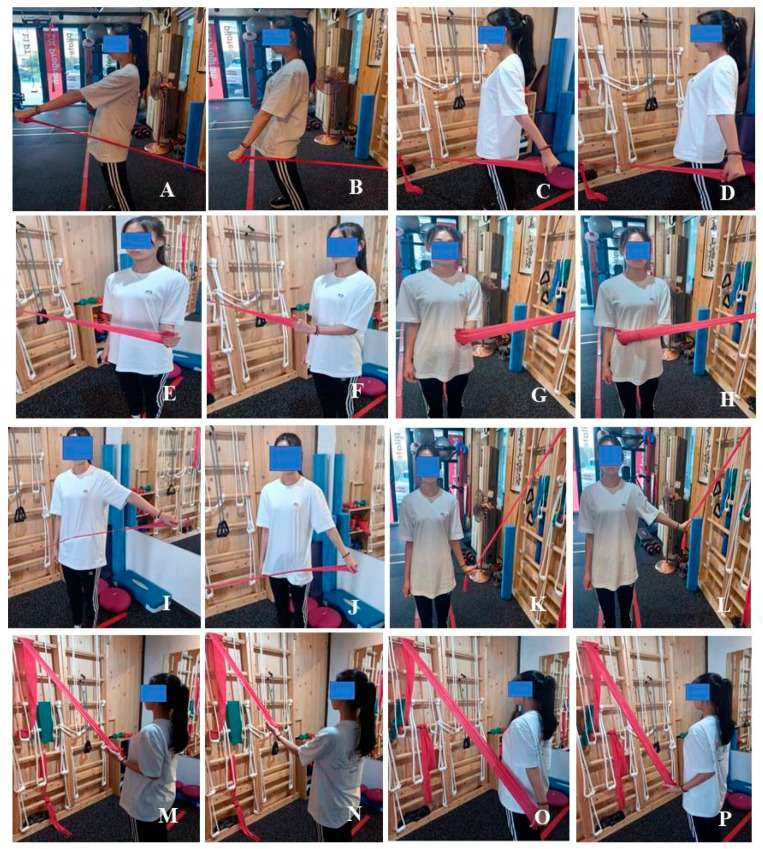
Exercise intervention: (**A**) forward flexion with concentric contraction; (**B**) forward flexion with eccentric contraction; (**C**) extension with concentric contraction; (**D**) extension with eccentric contraction; (**E**) external rotation with concentric contraction; (**F**) external rotation with eccentric contraction; (**G**) internal rotation with concentric contraction; (**H**) internal rotation with eccentric contraction; (**I**) abduction with concentric contraction; (**J**) abduction with eccentric contraction; (**K**) adduction with concentric contraction; (**L**) adduction with eccentric contraction; (**M**) rowing with concentric contraction; (**N**) rowing with eccentric contraction; (**O**) rowing plus with concentric contraction; (**P**) rowing plus with eccentric contraction.

**Figure 3 ijerph-18-13078-f003:**
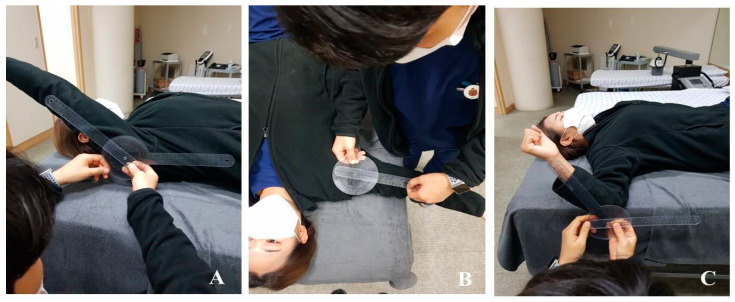
Measurement of the shoulder range of motion: (**A**) forward flexion; (**B**) abduction; (**C**) external rotation.

**Figure 4 ijerph-18-13078-f004:**
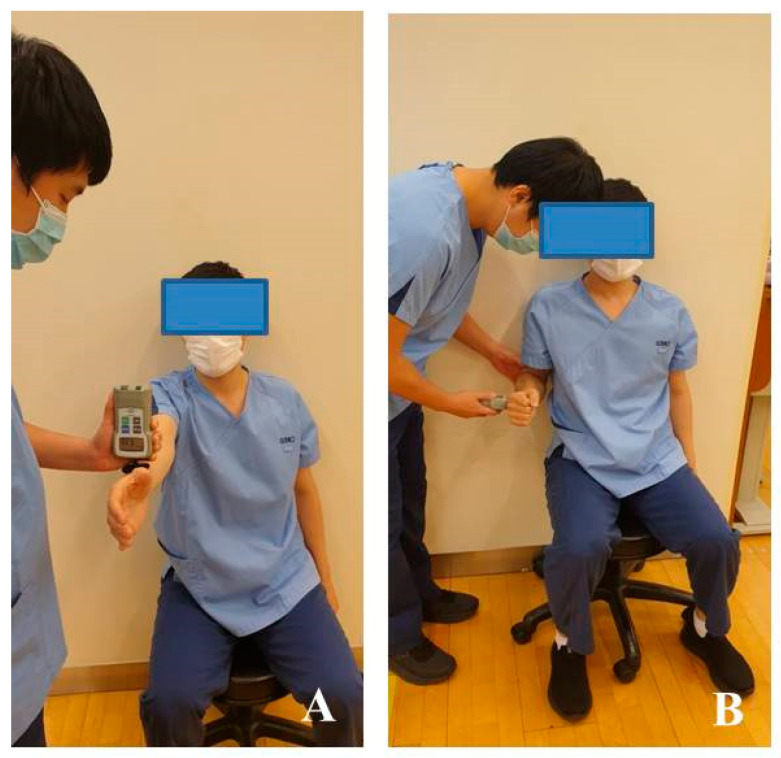
Measurement of muscle strength: (**A**) forward flexion; (**B**) external rotation.

**Table 1 ijerph-18-13078-t001:** Characteristics of the participants in each group.

Variables	ECG (*n* = 15)	CCG (*n* = 15)	*p*-Value
Age (years)	51.53 ± 4.73	52.40 ± 4.03	0.594
Height (cm)	157.36 ± 4.49	160.00 ± 4.58	0.122
Weight (kg)	58.96 ± 4.79	61.15 ± 5.39	0.249
BMI (kg/m^2^)	25.28 ± 1.67	25.03 ± 1.40	0.664
VAS	7.27 ± 0.70	7.40 ± 0.51	0.556
Involved site (right:left)	7:8	7:8	-
ROM in flexion (°)	109.13 ± 10.77	109.73 ± 9.08	0.870
ROM in abduction (°)	60.20 ± 9.92	60.87 ± 7.02	0.833
ROM in ER (°)	25.53 ± 3.87	25.13 ± 3.50	0.769
Strength in flexion (kg)	6.83 ± 0.65	6.80 ± 0.66	0.912
Strength in ER (kg)	9.21 ± 0.92	9.50 ± 0.86	0.375
Constant–Murley score	35.67 ± 1.72	34.67 ± 3.18	0.298

Data are presented as mean ± standard deviation. BMI: body mass index; ECG: eccentric contraction exercise group; CCG: concentric contraction exercise group; ROM: range of motion; ER: external rotation; VAS: visual analog scale.

**Table 2 ijerph-18-13078-t002:** The detailed information about the main exercise in the exercise program.

Variables	ECG (*n* = 15)	CCG (*n* = 15)
Exercise frequency	3 times/week	3 times/week
Exercise intensity	VAS 3–4	VAS 3–4
Exercise time	60 min	60 min
Contraction time	Concentric contraction: 2 s Eccentric contraction: 10–15 s	Concentric contraction: 2 s Eccentric contraction: 3 s
Exercise type	Forward flexion, extension, ER, IR, abduction, adduction, rowing, rowing plus	Forward flexion, extension, ER, IR, abduction, adduction, rowing, rowing plus
Maximal exercise volume	15 reps, 3 sets	15 repetition, 3 Sets
Exercise progression	Initial reps: 10 reps Increase until 15 reps	Initial reps: 10 reps Increase until 15 reps

ECG: eccentric contraction exercise group; CCG: concentric contraction exercise group; VAS: visual analog scale; ER: external rotation; Rep: repetitions.

**Table 3 ijerph-18-13078-t003:** Clinical outcomes after an exercise intervention between the groups.

Variables	Group	Pre	Post	*t*	Δ%		*F*-Value	*p*-Value
VAS	ECG	7.27 ± 0.70	2.87 ± 0.64	20.579 ***	−61	T	749.75	<0.001
						G	12.064	0.002 **
	CCG	7.40 ± 0.51	3.93 ± 0.59	18.065 ***	−47			
						T × G	10.554	0.003 **
ROM in flexion (°)	ECG	109.13 ± 10.77	157.20 ± 11.73	−12.019 ***	31	T	152.370	<0.001
						G	1.995	0.021 *
	CCG	109.73 ± 9.08	141.27 ± 16.40	−6.234 ***	22			
						T × G	6.573	0.016 *
ROM in abduction (°)	ECG	60.20 ± 9.92	123.67 ± 7.67	−20.081 ***	51	T	712.549	<0.001
						G	1.720	0.200
	CCG	60.87 ± 7.02	116.07 ± 12.30	−17.657 ***	48			
						T × G	3.458	0.073
ROM in ER (°)	ECG	25.53 ± 3.87	56.00 ± 10.47	−11.179 ***	54	T	255.584	<0.001
						G	4.235	0.049 *
	CCG	25.13 ± 3.50	49.27 ± 6.37	−11.725 ***	49			
						T × G	3.439	0.074
Strength in flexion (kg)	ECG	6.83 ± 0.65	9.09 ± 0.93	−17.466 ***	25	T	326.767	<0.001
						G	1.503	0.230
	CCG	6.80 ± 0.66	8.52 ± 0.63	−9.655 ***	20			
						T × G	6.015	0.021 *
Strength in ER (kg)	ECG	9.21 ± 0.92	12.67 ± 0.90	−22.664 ***	27	T	813.889	<0.001
						G	4.726	0.038 *
	CCG	9.50 ± 0.86	10.99 ± 0.96	−18.096 ***	14			
						T × G	130.047	<0.001
Constant–Murley score	ECG	35.67 ± 1.72	69.20 ± 5.31	−23.121 ***	48	T	577.340	<0.001
						G	9.515	0.005 **
	CCG	34.67 ± 3.18	62.13 ± 7.64	−13.182 ***	44			
						T × G	5.710	0.024 *

Data are presented as mean ± standard deviation. ECG: eccentric contraction exercise group; CCG: concentric contraction exercise group; ROM: range of motion; ER: external rotation; VAS: visual analog scale. *F*: the results for the two-way repeated measure ANOVA, *t*: the results for the paired *t*-test. * *p* < 0.05, ** *p* < 0.01, *** *p* < 0.001.

## Data Availability

No new data were created or analyzed in this study. Data sharing is not applicable to this article.

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
