# Peer review of "Effects of Different Types of Contraction Exercises on Shoulder Function and Muscle Strength in Patients with Adhesive Capsulitis"

_ijerph, 2021, doi:10.3390/ijerph182413078_

Round 1
Reviewer 1 Report
The paper by Kim et al. concerns the effect of eccentric versus concentric training as therapy in female patients with adhesive capsulitis of the shoulder syndrome (ACS)..
The aetiology of ACS is not clearly understood and there remains a lack of consensus in the clinical management of this condition. This syndrome can in fact be either primary or secondary. The primary form occurs in patients presenting with painful and restricted shoulder movements in which no underlying cause is found; secondary causes of ACS include trauma, diabetes, thyroid disease and other autoimmune disorders.
The data presented in the paper under review are quite innovative, formally statistically correct, and clearly show the different effect induced in patients by the two physical activity trainings used indicating that the treatment based on eccentric contraction training is more effective in increasing the functional capacity of the shoulder using the indicated parameters such as range of motion (ROM), muscle strength and muscle function measurements.
As mentioned at the end of the discussion, the authors recall that this syndrome is characterised by different clinical phases (only three according to the authors, four according to the most recent literature.) to be used as a useful method to monitor and evaluate the progression of the symptoms (painfull , freezing, frozen and thawing), during the period of 1-2 years necessary for the functional restoration of the shoulder (Date A, Rahman L. Frozen shoulder: overview of clinical presentation and review of the current evidence base for management strategies. Future Sci OA. 2020;6(10):FSO647. Published 2020 Oct 30. doi:10.2144/fsoa-2020-0145).
From what you can read in the Discussion section, (lines 249-50), the recruitment of patients was quite correct as all the selected subjects did so. Even though stages I and II of the syndrome are not specified, this does not significantly affect the comparative homogeneity of the statistical sample since the subjects recruited belong to the early stage of AC.
In order to eliminate any possible doubt and as the first suggestion that I would like to give to the authors, it is to specify better in the Methods section, the clinical and etiopathogenetic aspects of the two groups analysed (also using a table).
Another aspect that needs to be discussed is the knowledge of an important parameter to understand if the training procedure used is really effective against ACS. This parameter concerns the effect that the training program has on the recovery time of the patients involved.
In my opinion, it is important, in order to establish the correct use of this treatment in patients with ACS, to know the effects it has as a function of time of application and not only after 12 weeks. Some debilitating situations may be involved earlier than others (e.g. pain and/or ROM) and this information may be very important in determining the treatment to be proposed (including the use of other therapeutic strategies).
If the authors have data available after different training periods (e.g. after 3, 6, 9 weeks) of the measurements taken, could add a graph in the paper to highlight this. This would be very useful and would, in my opinion, increase the applicative value of the paper.
Reviewer 2 Report
Thank you for the opportunity to review this manuscript. The purpose of the study was the comparison of an eccentric contraction exercise group to a concentric contraction exercise group in individuals with adhesive capsulitis.
I completely see the interest of the topic for the readership of the scope of IJERPH.
However, I want to address some major comments that should help increasing the quality of the manuscript prior to potential publication.
Abstract/Title:
Generally, the Abstract covers all necessary content.
However, please, grammatically revise the first sentence of the Abstract (L19-20).
L30-32: I am hesitant that this statement is generally supported by one intervention study on two groups with a sample size of 15 without any control group that did not execute any training program. I would be more hesitant with concluding that in a more study-based manner
Keywords:
I would recommend more powerful keywords that better highlight the scope of your study.
Manuscript:
Introduction:
General comments:
Generally, the introduction covers all necessary content deduce the purpose of the study. However, I miss the clear elaboration of your research questions.
Specific Comments:
L42-43: Please revise sentence for understandability.
L43-45: Although giving us the respective source, I would recommend describing the three stages of AC in a sentence for a better distinction between those stages.
L45-46: Completely revise sentence on grammatical level.
L49 et seq.: You’re reporting here quite some studies that investigated effects of training therapy on AC. In the beginning of your Abstract, you’re reporting that there is a lack of studies. With that two ambivalent exclamation you’re contradicting yourself. Please modify for better structure of the manuscript.
L52-54: In that section, you’re reporting even more studies.
L54 et seq.: I would suggest that you better elaborate that section in terms of clear and logic way that leads to your research questions and hypotheses. Accordingly, you’re telling your aim or purpose of the study, but clear scientific hypotheses or research questions are mandatory at the end of the Introduction, due to the research gap that you detected based on your findings from combing the literature.
Methods:
Specific:
L70-71: What’s the reasoning for defining this effect size and the power? Did you take those from studies in the same field? Did you do a pilot before-hand?
L69-72: This sentence also needs grammatical revision.
L72-74: You need to give more details on your study groups. At least, anthropometrics as age, height, and weight. Furthermore, how did you randomize? And, did you care that the groups are homogenous according to the anthropometric data or did they differ in some factors? If so, this could clearly influence the outcome. You did that in the Results section. However, the Table appears totally disconnected there.
L74-76: And what had been the inclusion criteria. How did you grade different stages of the groups? Did the groups differ overall in the symptoms according to the disease? This is a potential serious flaw of your manuscript. Please, thoroughly revise this part. Especially, in intervention studies it is of strong importance to care for the correct set-up of the study groups.
L78-80: Please, grammatically revise. In English there is always the place before the time.
L81: I insist to structure the Methods different: the exercise program has to be presented before the outcome measures are described.
Section 2.2.1: Can you give a figure how the goniometer was attached at the shoulder. Your explanation is fine but it would be greatly supported by a picture.
L97-98: If you want to have this statement like this in your manuscript, you need to give references for that, here. If it is used that commonly you need to cite other studies.
Furthermore, grammatical revision.
2.2.1/2.2.2/2.2.3 I would arrange the three ‘tests’ in a different order. First, the scoring system, second the RoM test, and third the strength test. This is a more logic order and rather the order in which you conducted those through the experiment.
2.2.3 I would recommend a figure for this procedure as well.
2.3. Give me more explanations on the training bouts. So, how many trials did the individuals perform overall. Was this controlled for all participants? How did you control standardization of the training? Did the participants perform any other sports aside during the study participation?
It’s not sufficient to just mention in the text ‘Both groups performed istonic exercise,…’ and then you just generally describe what they had done and then, in the figure, you’re extensively describe the exercises. Well, at least there has to be a mentioning of the performed exercises in the manuscript, comparable to the caption of Figure 1. However, the general movement characteristic would be sufficient. Kind of: exercises for anteversion and retroversion of the shoulder and so on.
I’m generally not convinced if you could answer your research question without having a control group that did not execute any kind of training. This is a very serious flaw of your study design. You should bring up some strong arguments for the decision-making reasons. It is just standard good scientific work to have always a Null Control group in intervention studies.
Results:
General: Could you structure your Results section clearer? It is hard to read, because you mention that many results. Maybe it would be helpful that you create sub-headings that it is clearer, especially, about which movement execution that results are based on. Or at least make some sub-sections.
L144-153: That’s all Methods from my point of view, including Figure 2 and Table 1.
L158-160: These are lots results in one sentence. Could you disentangle this part, please?
L164-165: I would be hesitant in reporting unsignificant results. Maybe, at all, concentrate on the results to report that are mandatory to answer your research questions.
L168: VAS results should come before the results of biomechanical analysis variables, e.g., flexion RoM results. Build your Results section on the same structure as you reported/described the variables/outcome measures in the Methods section. And remind that I already had some comments on the structure as well. You have to re-modify those parts, please.
L173-175: Grammatically revise sentence, please.
L187: This is the only occasion, where you’re referring to Table 2 in the text. However, aren’t the other parts of the Results section not also linked to Table 2. Refer the several sections to the Table. Or consider building smaller sub-tables that belong to each sub-result section. I believe this would strongly help to follow your results section.
Table 2: Reporting significance values of .000 is just no good presenting style. If you have quite low p-values, please report those as < .001.
Discussion and Conclusion:
At the beginning of the Discussion, it is mandatory to repeat the research questions or the purpose of your investigation.
L204-205: I do not see the link of that statement to the one before.
L205-209: This is rather conclusive statement about your findings and potential further investigation. Afterwards, L210, you start again with description of AC disease and so on. I would re-order the structure of the Discussion. At the end, you summarize your findings, tell us the limitations of your study and lead over to the Conclusions based on your findings.
Accordingly, also in the Discussion, exercise related analysis of pain should come at the beginning, before you give findings of RoM measures, and strength. Because, if I would have heavy pain in the shoulder, and I have shoulder arthrosis, automatically my shoulder RoM and as well my arm strength is reduced.
L228: Give the reference at the correct position. Here, it appears at if it’s belonging to the latter part of the sentence.
The Discussion generally lacks of a clear discussion of the own findings in relation to the findings in literature, according to training program generally and specifically.
L248: Here, you’re reporting that you’re results have no aim for generalization. For the reasons you mention, I agree. However, in L 256-257, you stating a very general conclusive phrase, according to your finding. However, this Conclusion is not supported by your study.
L250-251: You should include those points in your Methods for better describing your sample. So, freezing stage was an inclusion criteria; I read that at first occasion here; please revise your inclusion and exclusion criteria.
Reviewer 3 Report
Thank you for the opportunity to review this paper. Clinical trials such as this one are of high value. I think the study was conducted very well in general. The language is also very good, the paper is easy and pleasant to read. Congratulations. However, I believe that the paper could be much, much stronger. Below, I provide recommendations for several improvements. The authors need to carefully address these points. Otherwise, I think I probably won’t be able to recommend this paper for publication.
- One of the most important points to address is the rationale for the study. In lines 59-60, you state several previous studies. How is your study different from these, what new knowledge does it bring? What did you do different or better than previous studies? Related to that, I also think that these studies (1,14,20,21) should be described more, with 1-2 sentences for each, so the reader is more informed regarding previous findings. Lastly, building on previous finings, you should add a hypothesis at the end.
- Line 75-76. Excluding patients with other pathologies, such as impingement syndrome, is important for the rigor of the study. However, the diagnostic procedure is not explained. How did you determine specific pathologies (provocative tests, diagnosis by doctor, or some other way?). This is another point that needs to be explained in detail. You refer to participants to being diagnosed in line 247, but the procedure is not explained.
- Hand-held dynamometry. Is the reference [24] also including the reliability that you report (r = 0.99). If that is so, then repeat the reference [24] after this part of the sentence as well. Otherwise, explain the origin of this 0.99 score. It is suspiciously high to me. Moreover, it is not clear if ‘r’ represents Pearson’s r, or ICC?
- Section 2.3 should include more details. Was the intensity of the exercise standardize/individualized? You only say you monitored it with VAS, but how did you choose an intensity? If the intensity was always set to match VAS 3-4, this should be state more explicitly. Moreover, was it progressed in any way along the duration of the study? How many sets of each exercise? Was the VAS scale 1-10 or 0-10? Were the scores reported by just saying the pain level, or did you use actual physical VAS scales? Also, what exactly was the question you asked the participants for this assessment?
Another point is the duration of ECG and CCG. You say ECG was 2 second/10-15 second, but for the CCG, you only report one value (3 s). What about the duration of CCG in CCG exercise?
- Why did you use separate t-tests for assessing group and time effect? The 2-way RM Anova will give you main time and main group effects. You can add t-tests then only to assess time effect in each group separately. Moreover, effect sizes should be added to the results, and the scale for their interpretation into statistics section. Moreover, explain on the beginning of the results, if the variables were normally distributed or not (provide a range of p-values for Smirnov test)
- Line 73-74. If I understand correctly, the participants were allocated as follows: 1,3,5,7… to one group and 2,4,6,8… to the other group. This is not really random, especially if the numbers represent the order of participation. Please elaborate, and change the wording in Figure 2 – instead of random division, add something like ‘’group allocation’’.
- The results should include effect sizes (text) and % changes in individual groups (tables). Consider integrating these also into the abstract.
- The discussion is somewhat vague. You should discuss the potential reasons for the greater effects of ECG over CCG. You mention higher motor unit recruitment, but this is wrong – in you case, the force is the same for the two conditions (at least that is what it should be the case, as you used elastics). For the same force, the ECG should be characterized by lower activation and motor unit firing rate.
- The comparison of you results to previous studies should also be more detailed, with % changes reported for each, and the speculating why one or the other was more effective. Do not just say ‘’there were differences in populations and exercise programs’’, be specific and try to discuss in detail. At the end, you should emphasize what did you study add to the existing evidence.
Minor comments:
- Abstract: state that participants were female
- Abstract: add basic demographic participant data
- I think % changes and/or effect sizes and/or Means with SDs would be much more informative to the reader.
- Line 70: On what was the 0.25 value based on?
- Line 85: knees (plural)?
- 2.1 – what were the instructions to the participants. I know it is self-evident to people who know well how to perform active flexibility test, but it would be good to include what the instruction to the participant was.
- Figure 2 – if possible, provide reasons for withdrawal of 4 participants
- Lines 210-212. This is not necessary in the discussion; it is just repeating the intro. Discussion should be reserved for contrasting you results to previous findings.
Round 2
Reviewer 2 Report
Dear authors, thanks for addressing all risen points.
Some lingering minor comments remain that you please address before full acceptance of your manuscript.
Abstract/Title:
L35-37: please grammatically revise sentence.
Keywords: Please capsulitis instead of capsulitides as this term is more common.
Consider to add two more keywords to gain more visbility.
Keywords:
- Point: I would recommend more powerful keywords that better highlight the scope of your study.
Response: According to your comment, we changed the key words to others using MeSH terms. It was shoulder adhesive capsulitides, muscle contraction, and exercise therapy.
Manuscript:
Introduction:
Specific Comments:
L49-52: There are some minor grammatical errors.
L66: Please use correct referencing. I.e., Guler-Uysal et al. (year) have conducted …. [14].
L83-84: There is still a gap between the unraveled studies and your deduced study purpose. You need to define a clear research gap after unraveling the studies. What did they found that leads to your study purpose now? You need to summarize their findings, before stating that the aim of your study was…
Methods:
Specific:
L87-90: This is still not addressed properly. How defining such an effect size and such a power of your study? You cannot just arbitrarily take any values without any rationale.
Thanks for modification of Figure 2 and the addition of Table 2. As well as Figure 3 and 4. Good job. Way better to understand what you did.
Results:
General: These are extremely high Cohen’s d effect size. How do you explain that? Furthermore, please finely text-edit the results section that your p-values are reported correctly (e.g., p=074(?)). What does that mean. There is a punctuation missing.
Reviewer 3 Report
The authors did a very good job with the revision. I consider the paper acceptable for publication. Thank you and best regards,
Author Response
The author sincerely thank you for your effort for improving this study.